# Comparative Evaluation of Rapid Isothermal Amplification and Antigen Assays for Screening Testing of SARS-CoV-2

**DOI:** 10.3390/v14030468

**Published:** 2022-02-25

**Authors:** Nol Salcedo, Brena F. Sena, Xiying Qu, Bobby Brooke Herrera

**Affiliations:** 1E25Bio, Inc., Cambridge, MA 01239, USA; nsalcedo@e25bio.com (N.S.); brena@e25bio.com (B.F.S.); xiyingqu66@gmail.com (X.Q.); 2Department of Immunology and Infectious Diseases, Harvard T.H. Chan School of Public Health, Boston, MA 02115, USA

**Keywords:** SARS-CoV-2, surveillance, screening, testing, isothermal molecular test, antigen test

## Abstract

Human transmission of SARS-CoV-2 and emergent variants of concern continue to occur globally, despite mass vaccination campaigns. Public health strategies to reduce virus spread should therefore rely, in part, on frequent screening with rapid, inexpensive, and sensitive tests. We evaluated two digitally integrated rapid tests and assessed their performance using stored nasal swab specimens collected from individuals with or without COVID-19. An isothermal amplification assay combined with a lateral flow test had a limit of detection of 10 RNA copies per reaction, and a positive percent agreement (PPA)/negative percent agreement (NPA) during the asymptomatic and symptomatic phases of 100%/100% and 95.83/100%, respectively. Comparatively, an antigen-based lateral flow test had a limit of detection of 30,000 copies and a PPA/NPA during the asymptomatic and symptomatic phases of 82.86%/98.68% and 91.67/100%, respectively. Both the isothermal amplification and antigen-based lateral flow tests had optimized detection of SARS-CoV-2 during the peak period of transmission; however, the antigen-based test had reduced sensitivity in clinical samples with qPCR Ct values greater than 29.8. Low-cost, high-throughput screening enabled by isothermal amplification or antigen-based techniques have value for outbreak control.

## 1. Introduction

Alongside widespread vaccine campaigns, strategies continue to be implemented to reduce the human transmission of severe acute respiratory syndrome coronavirus 2 (SARS-CoV-2) [1,2,3,4]. Testing, in particular, has played an important role throughout the coronavirus disease 2019 (COVID-19) pandemic in detecting the virus and emergent variants of concern, enabling responses at the national, community, and individual levels [5,6]. However, most testing occurs in centralized settings that utilize quantitative polymerase chain reaction (qPCR) assays [7,8,9]. While these molecular techniques can detect minute amounts of viral RNA and therefore are most appropriate for clinical diagnosis, they cannot be scaled to meet demands for extensive public health surveillance or frequent screening of individuals, especially in resource-limited settings. Inexpensive, accurate tests that can be self-administered or performed at the point-of-care, and provide actionable results, will further facilitate outbreak suppression.

Diagnostic testing for COVID-19 focuses on establishing the presence or absence of SARS-CoV-2 in symptomatic or asymptomatic individuals [9,10]. In general, healthcare professionals collect respiratory specimens via nasopharyngeal swabs or use less invasive approaches such as anterior nares swabs or saliva collection [11]. The respiratory specimens then are processed by centralized high-complexity laboratories with specialized equipment using qPCR assays with results being reported within 24 to 48 h. In some regions, bottlenecks in laboratory-based testing have led to turnaround times exceeding several days, diminishing the efficacy of this approach to prevent ongoing transmission.

Surveillance testing estimates the infection dynamics of SARS-CoV-2 in representative sample sets. Molecular-based techniques that are highly sensitive and specific, and report qPCR cycle thresholds (Ct) and viral loads, are typically used for surveillance testing [12,13,14]. An emerging approach involves surveillance of SARS-CoV-2 via wastewater monitoring using qPCR assays [15,16]. In general, surveillance testing is performed in centralized settings with the resulting information used to monitor epidemic trajectory in specific communities and allow for real-time evaluation and/or implementation of mitigation programs. Additionally, surveillance testing can be conducted with antigen tests using de-identified and/or pooled nasal swab specimens. A proof-of-concept study demonstrated that an antigen test can detect SARS-CoV-2 positive nasal swab specimens up to Ct value 30.1 (viral load 4.8E4 viruses/mL), even when the positive specimen is pooled into negative nasal swab specimens [17].

Identification of individuals who are likely infectious with screening testing is one of the most effective, but underused, strategies to limit the ongoing transmission of SARS-CoV-2 [18]. In approximately 20–40% of COVID-19 cases, the infection remains asymptomatic, and symptomatic disease is preceded by a pre-symptomatic incubation period [19,20,21]. Yet, pre-symptomatic and asymptomatic cases contribute significantly to the SARS-CoV-2 spread, challenging our ability to contain outbreaks [9,19].

Breaks in transmission chains can be most effectively achieved when screening testing is applied frequently and serially using self-administered rapid tests [22,23,24,25,26]. Antigen-based tests, which utilize combinations of monoclonal antibodies and nanoparticles to detect viral proteins, do not require instruments or skilled operators; as of December 2021, 10 antigen tests for SARS-CoV-2 were approved for at-home use in the United States [27]. Although antigen tests have lower analytical sensitivity and specificity compared to qPCR assays, they have increased ability to detect SARS-CoV-2 during the acute phase of COVID-19 when an infected individual is most likely to transmit the virus [17,19,28,29].

Moreover, isothermal amplification technologies offer the simplicity and speed of antigen tests but have higher sensitivity and specificity [30,31,32,33,34,35]. One of the most promising isothermal amplification technologies is recombinase polymerase amplification (RPA) [33,34]. In RPA, double stranded DNA denaturation and strand invasion is achieved by a cocktail of enzymes including recombinases, single-stranded binding proteins, and DNA polymerases; typically, this occurs by multiple heat cycles in PCR [35]. RPA has added benefits over other isothermal amplification technologies (i.e., loop-mediated isothermal amplification, LAMP, or CRISPR) as reactions occur at ambient temperatures (37–42 °C), in shorter time periods, and with results that can be visualized on a lateral flow test. One of three isothermal amplification technologies currently available in the United States for at-home detection of SARS-CoV-2 utilizes reverse transcription RPA (RT-RPA), and has been shown to detect the virus in nasal swab specimens with as low as twenty genome copies [36]. Given their robust sensitivity and specificity, RT-RPA assays are optimized to detect SARS-CoV-2 during the peak period of transmission in individuals with pre-symptomatic, symptomatic, and/or asymptomatic infections [33,34].

Here, we performed a comparative evaluation of a RT-RPA assay and an antigen test. Using previously characterized nasal swab dilution specimens, we assessed the analytical sensitivity of the two tests. We show that the RT-RPA assay allows for detection of SARS-CoV-2 down to 10 RNA copies per reaction compared to folds higher with the antigen test. We then calculated the positive percent agreement (PPA, or sensitivity) and negative percent agreement (NPA, or specificity) using stored, unextracted nasal swab specimens collected from individuals with or without COVID-19. We demonstrate that the RT-RPA assay has increased sensitivity in nasal swab specimens, particularly in qPCR Ct values greater than 29.8, regardless of if the sample was collected during the asymptomatic or symptomatic phases. Supporting the innovation, manufacturing, approval, and distribution of isothermal amplification screening tests will enable more effective control of infectious disease outbreaks.

## 2. Materials and Methods

### 2.1. Clinical Samples

The nasal swab dilution panel was provided by the non-profit PATH (www.path.org, accessed on 24 March 2021). Nasal swab dilutions were prepared from human nasal swab eluate discards from suspected COVID-19 patients, collected within seven days of post-symptoms onset. A single swab eluate positive for SARS-CoV-2 by qPCR was diluted into a single nasal eluate negative for SARS-CoV-2 by qPCR. For the dilution specimens with lower than 5000 RNA copies, known quantities of RNA (ATCC, Manassas, VA, USA) were spiked into nasal eluates negative for SARS-CoV-2. Dilution specimens were de-identified, coded, and then aliquoted and frozen at −80 °C. Aliquots were thawed and characterized by qPCR as previously described [17]. The primary studies under which the samples were collected received ethical clearance from the PATH Institutional Review Board (IRB) (approval number 0004244).

Additionally, nasal swab specimens were collected from a cohort of suspected COVID-19 patients with or without symptoms at a point-of-care site (POC nasal swab specimens); for individuals with symptoms, specimens were collected within the first 3 days of symptoms’ onset. The nasal swabs were mixed in tubes containing 1X PBS (MilliporeSigma, Burlington, MA, USA). Aliquots were de-identified, coded, and then frozen at −80 °C. The study under which the samples were collected received ethical clearance from the Advarra, Inc. IRB (approval number Pro00044496).

### 2.2. qPCR

Quantities of 200 μL of the POC nasal swab specimens were used for extraction with the MagMAX Viral/Pathogen II Nucleic Acid Isolation Kit (ThermoFisher Scientific, Waltham, MA, USA) on an epMotion 5075 (Eppendorf, Hamburg, Germany) liquid handler. Nucleic acids were eluted in 50 μL; 2 μL were used for qPCR confirmation using the GoTaq Probe 1-Step RT-qPCR System (Promega, Madison, WI, USA) on a QuantStudio 7 Flex Real-Time PCR Instrument (ThermoFisher Scientific, Waltham, MA, USA). The SARS-CoV-2 (2019-nCoV) CDC qPCR Probe Assay was used to detect the human RNaseP gene and two viral targets, 2019-nCoV_N1 and 2019-nCoV_N2 (Integrated DNA Technologies, Coralville, IA, USA).

### 2.3. RT-RPA Assay

Prior to isothermal amplification testing, the nasal swab dilution specimens or POC nasal swab specimens were lysed at 95 °C using a heat block (Southern Labware, Cumming, GA, USA) for 3 min. Isothermal amplification reactions were conducted using AmpliFast enzymes and a buffer (E25Bio, Inc., Cambridge, MA, USA), 1 μL RNase H (5U/μL; ThermoFisher Scientific, Waltham, MA, USA), 0.5 μL SuperScript IV RT (200 U/μL; ThermoFisher Scientific, Waltham, MA, USA), 0.5 μL of SARS-CoV-2 nucleocapsid (N) forward and reverse primers (300 nM final concentration), and 2 μL input template (nasal swab dilution specimen or POC nasal swab specimen). This mix was activated by the addition of 1 μL of magnesium acetate (14 nM final concentration), MilliporeSigma, (Burlington, MA, USA) followed by thorough mixing. Reactions were incubated at 38 °C for 20 min. A hybridization mix was prepared by combining 1 μL SARS-CoV-2 N biotinylated probe (0.167 nM final concentration) with 19 μL Tris pH 8 (10 mM). A total of 20 μL of the hybridization mix was added to each reaction, and samples were heated to 95 °C for 3 min followed by a cooling step at room temperature for 3 min. Quantities of 40 μL of buffer (Pocket Diagnostic, York, UK) were added to each reaction; then, the mixture was applied to the PCRD nucleic acid lateral flow test (Pocket Diagnostic, York, UK) and allowed to react for 10 min. Interactions of the immobilized test and control line antibodies with amplified nucleic acids and the nanoparticle conjugate produced visible bands, indicating whether a test was positive or negative.

### 2.4. Antigen Test

Rapid antigen tests (E25Bio, Inc., Cambridge, MA, USA) contain a monoclonal antibody and a nanoparticle conjugate that detect SARS-CoV-2 N. Quantities of 100 μL of the nasal swab dilution specimens or POC nasal swab specimens were applied to the antigen test and allowed to react for 15 min. Interactions of the immobilized test and control line antibodies with the antigen and the nanoparticle conjugate produced visible bands, indicating whether a test was positive or negative.

### 2.5. Image Analysis

Results from the isothermal amplification lateral flow tests and antigen-based lateral flow tests were captured via the Passport App (currently available through Apple, Inc.’s TestFlight; E25Bio, Inc., Cambridge, MA, USA). The images were machine-read and processed to quantify test results. The average pixel intensity was quantified at the test line, control line, and background areas. The background-subtracted test line signal was then normalized to the background-subtracted control line signal, and the final test signal was expressed as percent of control. The Passport App only stores images and identifiable test results locally on the user’s mobile device, and the individual can share the results with whomever, whenever they choose.

### 2.6. Statistics

GraphPad Prism 9.0 (San Diego, CA, USA) was used to analyze and report the performance of the isothermal amplification and antigen tests compared to qPCR. The sensitivity was defined as the fraction of total qPCR confirmed positive samples that are true positives according to the test. The specificity was defined as the fraction of total qPCR confirmed negative samples that are true negatives according to the test. Sensitivity and specificity calculations were based on a per-patient basis. Where appropriate, test signals were plotted using symbol and line graphs according to asymptomatic or symptomatic infection status and qPCR Ct thresholds.

## 3. Results

In the RT-RPA assay, viral RNA is first copied to cDNA by reverse transcriptase, then degraded by RNase H. The cDNA product is amplified by RPA using a forward and a FAM-labeled reverse pair of primers specific to the target sequence. The amplified nucleic acid target is denatured and hybridized to a biotinylated probe. Dual FAM-labeled and biotin-labeled products are then detected on a lateral flow test that contains nanoparticles and detection molecules (i.e., anti-FAM antibody and streptavidin) specific for FAM and biotin (Figure 1A). Ιn the antigen test, the interaction of antibodies and nanoparticles with protein targets produces detectable bands (Figure 1B). Both the RT-RPA assay and the antigen test used in this study target SARS-CoV-2 N. To reduce errors in user-based interpretation, we used a mobile phone application to machine-read and quantify the RT-RPA and antigen test results (Figure 1C). Mobile phone image processing allowed test users to obtain an objective analysis of their results, despite varied use conditions, and share data in real-time.

We evaluated the analytical sensitivity of the RT-RPA assay and the antigen test using well-characterized nasal swab dilution specimens. The dilution specimens contained SARS-CoV-2 RNA copies ranging from 1 (Ct value 39.6) to 200,000 (Ct value 25.2). Consistent with expectations from qPCR, the RT-RPA assay yielded positive results with an input of 10 RNA copies per reaction (Ct value 37.3) (Figure 2A,B). The antigen test reproducibly had detectable results with dilution specimens between 40,000 and 30,000 copies (Ct values 28.3 and 29.2, respectively) of SARS-CoV-2 (Figure 2C,D). The RT-RPA assay had a detection limit several orders of magnitude lower than the antigen test.

To evaluate the RT-RPA and antigen tests further, we compared their sensitivity and specificity using stored, unextracted nasal swab specimens collected from individuals with or without COVID-19. A total of 114 nasal swab specimens were negative and 59 were positive for SARS-CoV-2 by qPCR. Of the 114 negative specimens, 76 were collected from asymptomatic cases and 24 were collected from symptomatic cases. Of the 59 SARS-CoV-2 positive specimens, 35 and 24 were collected from asymptomatic or symptomatic cases, respectively. All 114 negative specimens were negative by the RT-RPA assay, regardless of symptoms, corresponding to a 100% specificity (Figure 3, Table 1 and Table 2). Only 1 of 76 negative specimens from asymptomatic cases was positive by the antigen test, corresponding to a 98.68% specificity (Figure 3A, Table 1). All negative specimens from 24 symptomatic cases were negative by the antigen test (100% specificity) (Figure 3B, Table 2). These results confirmed a low false positive rate for the RT-RPA assay and antigen test.

All 35 SARS-CoV-2 positive specimens from asymptomatic cases were positive by the RT-RPA assay, corresponding to a 100% sensitivity (Figure 3A, Table 1). Of the 24 positive specimens from the symptomatic phase, only 1 tested negative (95.83% sensitivity) (Figure 3B, Table 2). In contrast, the antigen test detected 29 out of 35 (82.86% sensitivity) asymptomatic SARS-CoV-2 positives and 22 out of 24 (91.67% sensitivity) symptomatic positives (Figure 3, Table 1 and Table 2). Of note, the sensitivity of the antigen test decreased significantly in Ct values greater than 30.1, while the sensitivity of the RT-RPA assay was maintained (Figure 3). Altogether, these results demonstrated that the true positive rate of the RT-RPA assay was much higher than the antigen test especially during the asymptomatic phase and particularly in specimens with higher Ct values.

## 4. Discussion

One of the most promising strategies aimed at SARS-CoV-2 outbreak suppression is the surveillance or screening of infectious individuals. This type of testing requires frequent and serial testing of large populations that can be self-administered or performed at the point-of-care in high-transmission settings (i.e., schools, workplaces, etc.). The primary goal of surveillance or screening testing is to achieve population-wide effects by breaking transmission chains through identification of cases, especially during the pre-symptomatic or asymptomatic phases [37,38,39].

Modeling studies demonstrated that frequent rapid testing of large populations, even with varied test accuracies, can help achieve herd effects thereby suppressing transmission of SARS-CoV-2 [22,23,40]. In Slovakia, ~80% of the population was screened for COVID-19 using antigen tests [41]. In a 2-week period, 50,000 cases were identified, and along with other public health measures (i.e., wearing masks, quarantining, etc.), the incidence was reduced by 82%. Further, at-home antigen testing was performed twice per week in a coworking environment in Cambridge, MA over a 6-month period [24]. In the case of a positive test, an individual would undergo a 10-day quarantine prior to returning to the workplace. Twice-weekly testing identified 15 individuals infected with SARS-CoV-2, with a test sensitivity of 96.2% on days 0–3 of symptoms. This testing strategy allowed the activities of the coworking sites to continue without pause. While frequent testing has been shown to reduce transmission of SARS-CoV-2, other challenges may arise, especially during times of high incidence rates, including limited supplies (i.e., nasal swabs) needed to perform the rapid tests, exhausted testing capacity within labs, and dilemmas with contact tracing and reporting. To our knowledge, screening testing using isothermal amplification techniques has not been extensively evaluated.

In this study, we performed a comparative evaluation of a RT-RPA assay and an antigen test for SARS-CoV-2. We tested the analytical sensitivity using a nasal swab dilution panel. The RT-RPA assay had a detection limit far lower than the antigen test. We then analyzed the performance of the tests using qPCR characterized nasal swab specimens collected from individuals with or without COVID-19. The RT-RPA assay had a high sensitivity (>95%) and specificity (100%) in specimens from asymptomatic or symptomatic cases. In contrast, the sensitivity of the antigen test during the asymptomatic phase was much lower at 82.86%, and especially with specimens that had Ct values greater than 30. A likely explanation is that during the asymptomatic phase, SARS-CoV-2 viremia has not peaked, resulting in reduced viral antigens in respiratory specimens. Additionally, Ct values > 30 typically appear later in the course of SARS-CoV-2 infection (i.e., 7 days after exposure), when the virus is being eliminated by the immune system, clearing antigen levels. In support of our hypotheses, the sensitivity of the antigen test increased to >90% in nasal swab specimens collected from symptomatic cases within 3 days of symptoms onset.

Future work should broaden the evaluation of isothermal amplification and antigen assays to additional settings, sample types, and disease states (i.e., pre-symptomatic phase). Performance testing on prospectively collected samples will further corroborate preliminary findings. Additionally, samples collected at specific timepoints during the infection cycle will help elucidate the robustness of RT-RPA throughout the course of COVID-19. As the RT-RPA assay used in this study uses an inexpensive water bath and heat block, there is a need for these types of assays to perform reactions with consumer-designed hardware that would allow for at-home or point-of-care testing. Optimizing reaction mixes can also help reduce the temperatures and time required for test processing. Finally, the mobile phone application used in this study lessens user error by interpreting the results via pre-designed algorithms. Additional open-source, low-cost methods for data capture and reporting are warranted.

Public health surveillance and screening requires rapid, inexpensive, and sensitive tests that can be scaled for frequent and serial testing in large numbers. Antigen tests and upcoming isothermal amplification assays fit these needs and could be scaled to millions of tests per day. Despite being shown to be highly effective at detecting infectious individuals, there are only a handful of rapid tests currently available for self-administration or at-home use in the United States. Even with approvals, these manufacturers have been unable to meet the scale and demand, leaving individuals without access to these valuable, inexpensive, rapid testing options. The support of manufacturing, rapid approval processes, and distribution of screening tests will help control COVID-19 outbreaks.

## Figures and Tables

**Figure 1 viruses-14-00468-f001:**
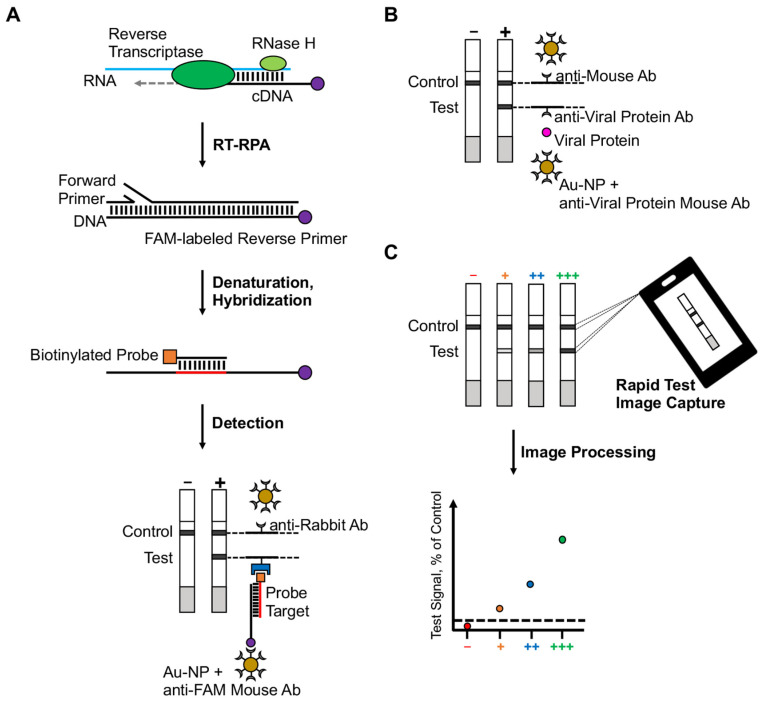
Schematic of RT-RPA assay versus antigen-based test. (**A**) In RT-RTPA, viral RNA is coped to cDNA by reverse transcriptase, then degraded by RNase H. Using a forward and a FAM-labeled reverse pair of primers specific to a target sequence, the cDNA product is amplified by RPA, then denatured and hybridized to a biotinylated probe. FAM-labeled and biotin-labeled products are detected on a lateral flow strip using molecules specific for FAM and biotin and nanoparticles. (**B**) In an antigen test, protein targets are detected by a lateral flow strip using protein-specific antibodies and nanoparticles. (**C**) A mobile phone application was used to image capture, machine-read, and quantify test results. The average pixel intensity is quantified at the test line, control line, and background areas. The background-subtracted test line signal is then normalized to the background-subtracted control line and expressed at % of control. − (red), test signal below the limit of detection; + (orange), low test signal; ++ (blue), medium test signal; +++ (green), high test signal.

**Figure 2 viruses-14-00468-f002:**
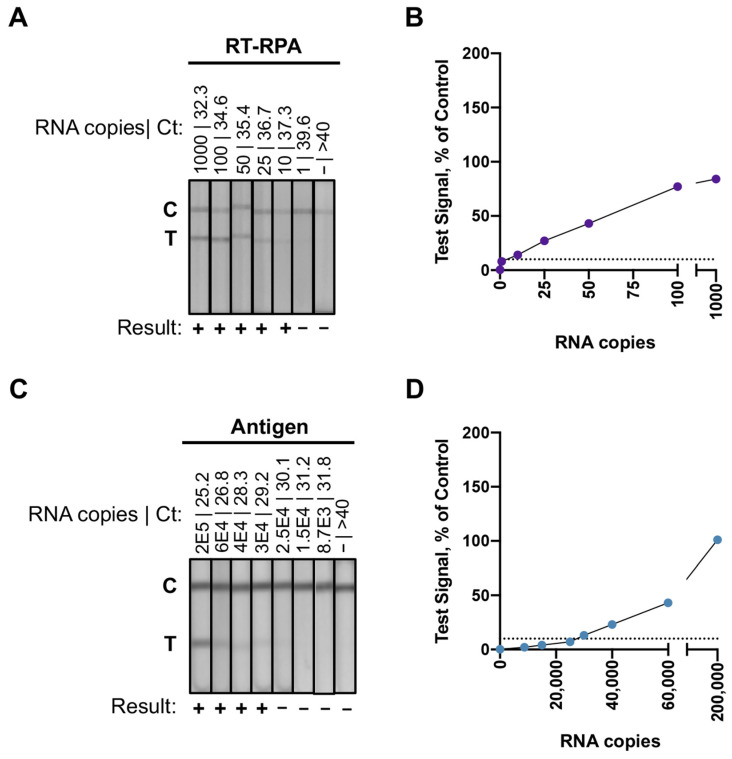
Analytical sensitivity of the RT-RPA assay and the antigen test using nasal swab dilution specimens. (**A**) Lateral flow strips for the RT-RPA reactions with dilution specimens containing RNA copies ranging from 0 to 1000. (**B**) Plot from the RT-RPA assay results quantified by the mobile phone application. The x-axis corresponds to dilutions’ specimens with known input copies of SARS-CoV-2 RNA. The y-axis corresponds to background subtracted test signal normalized to the control line for each lateral flow strip. Test results (purple dots) less than 10% of control are considered negative results, which is indicated by the black dashed line. (**C**) Lateral flow strips for the antigen tests with dilution specimens containing RNA copies ranging from 0 to 200,000. (**D**) Plot from the antigen tests results quantified by the mobile phone application. The x-axis corresponds to dilutions’ specimens with known input copies of SARS-CoV-2 RNA. The y-axis corresponds to background subtracted test signal normalized to the control line for each lateral flow strip. Test results (blue dots) less than 10% of control are considered negative results, which is indicated by the black dashed line.

**Figure 3 viruses-14-00468-f003:**
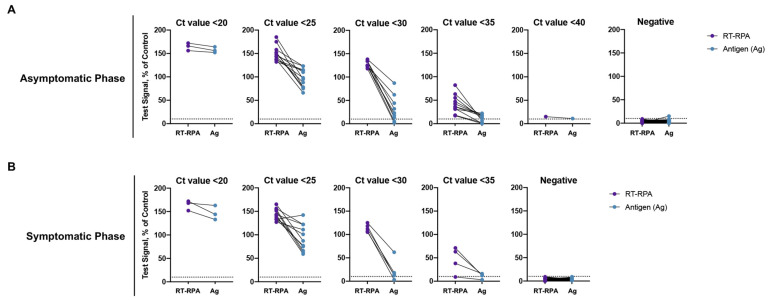
Clinical performance of the RT-RPA assay and the antigen test using nasal swab specimens collected from individuals with or without COVID-19. (**A**) Comparative evaluation of the RT-RPA assay (purple) and the antigen test (blue) using nasal swab specimens from asymptomatic cases. Comparative performance between the tests was plotted according to qPCR positive (Ct values between <20 to <40) and negative results. (**B**) Comparative evaluation of the RT-RPA assay (purple) and the antigen test (blue) using nasal swab specimens from symptomatic cases. Comparative performance between the tests was plotted according to qPCR positive (Ct values between <20 to <40) and negative results.

**Table 1 viruses-14-00468-t001:** Comparative performance of the RT-RPA assay and the antigen test against qPCR in asymptomatic cases.

Asymptomatic Phase								
		qPCR					95% CI	
		+	−	Total	PPA	100.00%	90.00%	100.00%
RT-RPA	+	35	0	35	NPA	100.00%	95.26%	100.00%
	−	0	76	76	PPV	100.00%		
	Total	35	76	111	NPV	100.00%		
		qPCR					95% CI	
		+	−	Total	PPA	82.86%	66.35%	93.44%
Antigen	+	29	1	30	NPA	98.68%	92.89%	99.97%
	−	6	75	81	PPV	96.67%	80.45%	99.51%
	Total	35	76	111	NPV	92.59%	85.78%	96.28%

**Table 2 viruses-14-00468-t002:** Comparative performance of the RT-RPA assay and the antigen test against qPCR in symptomatic cases.

Symptomatic Phase								
		qPCR					95% CI	
		+	−	Total	PPA	95.83%	78.88%	99.89%
RT-RPA	+	23	0	23	NPA	100.00%	90.75%	100.00%
	−	1	38	39	PPV	100.00%		
	Total	24	38	62	NPV	97.44%	84.80%	99.62%
		qPCR					95% CI	
		+	−	Total	PPA	91.67%	73.00%	98.97%
Antigen	+	22	0	22	NPA	100.00%	90.75%	100.00%
	−	2	38	40	PPV	100.00%		
	Total	24	38	62	NPV	95.00%	83.45%	98.62%

## Data Availability

All data produced in the present study are available upon reasonable request to the authors.

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
