# Peer review of "Comparative Evaluation of Rapid Isothermal Amplification and Antigen Assays for Screening Testing of SARS-CoV-2"

_viruses, 2022, doi:10.3390/v14030468_

Round 1

Reviewer 1 Report

In this manuscript, results of samples tested for SARS-CoV2 using multiple methods were compared to determine specificity and sensitivity of 2 rapid tests vs. the reverse transcription real time PCR. The manuscript is well written. Methods and results are explained clearly (Figure 1 is excellent for many levels of readers). No changes are requested.

Author Response

In this manuscript, results of samples tested for SARS-CoV2 using multiple methods were compared to determine specificity and sensitivity of 2 rapid tests vs. the reverse transcription real time PCR. The manuscript is well written. Methods and results are explained clearly (Figure 1 is excellent for many levels of readers). No changes are requested.

Response: Thank you for your comments. We appreciate you mentioning the manuscript, including its methods and results, were well written. Given the importance of helping all people understand research involving COVID-19, we also appreciate you mentioning that Figure 1 was comprehendible for readers of many levels. 

Reviewer 2 Report

The manuscript under review is an interesting work on the evaluation of two digitally integrated tests for SARS-CoV-2 to evaluate their performance. The Authors used nasal swabs previously collected from a cohort of suspected COVID-19 cases. They found that the antigen test was less sensitive (sensitivity < 82.86%) during the asymptomatic phase. The Authors provided a good discussion of their results. However, I think the manuscript could be implemented.

  • In the discussion, I suggest the Authors to add a paragraph on the real-life use and impact of the tests. In fact, they stated “Modeling studies have demonstrated that frequent rapid testing of large populations, 269 even with varied test accuracies, can help achieve herd effects thereby suppressing trans-270 mission of SARS-CoV-2”. However, during the worst phases of the pandemic, it was/is very difficult to perform frequent tests because of the huge number of infected people: no nasal swabs/test available in Pharmacies, overloaded laboratories for PCR, impossible contact tracing, ecc.
  • I suggest to implement the reference list with the following papers:

10.3390/ijerph18031268

10.1038/s41591-020-0891-7

10.3390/DIAGNOSTICS11091647
10.1016/j.ajic.2020.07.011

10.3390/ijerph17218033

Author Response

Response: Thank you for your comments. We appreciate you mentioning that we included a good discussion of our results. Per your suggestion, we have included in the discussion an additional statement with regard to some of the limitations that may arise with a frequent testing schema (lines 272-276). Finally, we have incorporated all of your suggested references into the manuscript.  

Reviewer 3 Report

Major issues:

1. For the estimation of sensitivity and specificity, please clarify that the calculation is based on a per-patient basis or a per-sample basis.

2. In line 228, the authors mentioned about a total of 173 nasal swab specimens including 114 nasal swab specimens were negative and 59 were positive for SARS-CoV-2 by qPCR. Please clarify these nasal swab specimens collected from how many individuals with or without symptoms.

3. In line 285-286, although the RT-RPA assay had a higher sensitivity (>95%) and specificity (100%) in specimens from asymptomatic or symptomatic cases, however, the cost is still relatively higher than that for antigen test. To promote the application of the RT-RPA assay, more convincing evidence and application fields should be provided.

4. The time point of sample collection is an important factor for the accuracy of screening test, how about this issue for those samples collected in this study?

5. In the “Discussion”, the authors mentioned about the mobile phone application used in this study lessens user error by interpreting the results via pre-designed algorithms. Please give a detail information in the “Materials and methods”.

Minor issue:

In addition to the RT-RPA assay, recent studies (Chen CC et al., 2021)with the diagnostic accuracy of antigen test are recommended to be discussed in this study. 

Author Response

Major issues:

1. For the estimation of sensitivity and specificity, please clarify that the calculation is based on a per-patient basis or a per-sample basis.

Response: Thank you for your comment. We have clarified in the methods section that sensitivity and specificity calculations were based on a per-patient basis (line 204).

2. In line 228, the authors mentioned about a total of 173 nasal swab specimens including 114 nasal swab specimens were negative and 59 were positive for SARS-CoV-2 by qPCR. Please clarify these nasal swab specimens collected from how many individuals with or without symptoms.

Response: Please refer to lines 235-237, which specifies that of 59 positive specimens, 35 were collected from asymptomatic cases and 24 were collected from symptomatic cases. 

3. In line 285-286, although the RT-RPA assay had a higher sensitivity (>95%) and specificity (100%) in specimens from asymptomatic or symptomatic cases, however, the cost is still relatively higher than that for antigen test. To promote the application of the RT-RPA assay, more convincing evidence and application fields should be provided.

Response: We appreciate your comments, however the emphasis of the study is not to "promote" RT-RPA assay. Rather, the study was conducted to evaluate the comparative results between antigen-based and isothermal amplification-based techniques. The present study included of 174 nasal swab specimens collected from individuals during both the asymptomatic and symptomatic phases of COVID-19. In addition to clinical testing, we performed a comparative evaluation of the antigen and isothermal amplification tests using a clinical dilution panel to determine limits of detection. Respectfully, we consider any additional clinical testing outside the scope of the study. Additionally, in the limitations section of the manuscript, we state that future work should broad the evaluation of isothermal amplification and antigen tests to additional settings, samples types, and disease states. 

4. The time point of sample collection is an important factor for the accuracy of screening test, how about this issue for those samples collected in this study?

Response: Thank you for your comment. We agree that the time point in which samples are collected impacts the accuracy of both antigen-based and isothermal amplification-based tests. While we did not have access to time point data, we do provide details as to whether the samples were collected during the asymptomatic or symptomatic phases and categorize these samples by the correspond RT-PCR cycle threshold values. Ct values roughly correlate with the timepoints in which samples are collected. In general, lower Ct values correspond to earlier timepoints, whereas high Ct values correspond to later timepoints within the infection cycle.  Further in the limitations section of the manuscript, we indicate that further evaluation should be performed at specific timepoints throughout the course of COVID-19 (lines 299-301). 

5. In the “Discussion”, the authors mentioned about the mobile phone application used in this study lessens user error by interpreting the results via pre-designed algorithms. Please give a detail information in the “Materials and methods”.

Response: Thank you for your comment. Please refer to lines 188-196, which describes mobile phone image capture of results and results interpretation. In brief, the mobile phone app calculates the test signal intensity and normalizes it to the control line signal. This algorithm ensures data consistency. 

Minor issue:

In addition to the RT-RPA assay, recent studies (Chen CC et al., 2021)with the diagnostic accuracy of antigen test are recommended to be discussed in this study. 

Response: Thank you for your comment. Per your suggestion, we have incorporated the reference Chen CC, 2021 into the manuscript. 

Round 2

Reviewer 3 Report

No more comments.